Effects of training frequency on muscular strength for trained men under volume matched conditions

Johnsen Emil
van den Tillaar Roland roland.v.tillaar@nord.no
Department of Sport Sciences and Physical Education, Nord University , Levanger , Norway
Guidetti Laura
Electronic publication date: 2021 Feb 18
Publication date: 2021
Volume: 9
Electronic Location ID: e10781
Received 2020 Aug 12; Accepted 2020 Dec 23
Copyright: © 2021 Johnsen and van den Tillaar
Copyright year: 2021
Copyright holder: Johnsen and van den Tillaar
License: This is an open access article distributed under the terms of the Creative Commons Attribution License, which permits unrestricted use, distribution, reproduction and adaptation in any medium and for any purpose provided that it is properly attributed. For attribution, the original author(s), title, publication source (PeerJ) and either DOI or URL of the article must be cited.
License URL: https://creativecommons.org/licenses/by/4.0/

Keywords: RPE, Rate of perceived exertion, Bench press, Squats

Funding: The authors received no funding for this work.

==============================
Background

In resistance training, the role of training frequency to increase maximal strength is often debated. However, the limited data available does not allow for clear training frequency “optimization” recommendations. The purpose of this study was to investigate the effects of training frequency on maximal muscular strength and rate of perceived exertion (RPE). The total weekly training volume was equally distributed between two and four sessions per muscle group.

Methods

Twenty-one experienced resistance-trained male subjects (height: 1.85 ± 0.06 m, body mass: 85.3 ± 12.3 kg, age: 27.6 ± 7.6 years) were tested prior to and after an 8-week training period in one-repetition maximum (1RM) barbell back squat and bench press. Subjects were randomly assigned to a SPLIT group (n = 10), in which there were two training sessions of squats and lower-body exercises and two training sessions of bench press and upper-body exercises, or a FULLBODY group (n = 11), in which four sessions with squats, bench press and supplementary exercises were conducted every session. In each session, the subjects rated their RPE after barbell back squat, bench press, and the full session.

Results

Both groups significantly increased 1RM strength in barbell back squat (SPLIT group: +13.25 kg; FULLBODY group: +14.31 kg) and bench press (SPLIT group: +7.75 kg; FULLBODY group: +8.86 kg) but training frequency did not affect this increase for squat (p = 0.640) or bench press (p = 0.431). Both groups showed a significant effect for time on RPE on all three measurements. The analyses showed only an interaction effect between groups on time for the RPE after the squat exercise (p = 0.002).

Conclusion

We conclude that there are no additional benefits of increasing the training frequency from two to four sessions under volume-equated conditions, but it could be favorable to spread the total training volume into several training bouts through the week to avoid potential increases in RPE, especially after the squat exercise.

Introduction

The interest in resistance training has risen in popularity (Wernbom, Augustsson & Thomee, 2007). Several studies pointed out that conducting resistance training had many potential health benefits for people of all ages (Winett & Carpinelli, 2001). An increase in overall strength through resistance training is also seen in the context of an increase among athletes in a variety of sports (Suchomel, Nimphius & Stone, 2016). Resistance training is an important factor in maintaining and developing muscle mass and muscle strength. To maximize these adaptations in human muscles, the manipulations of various resistance training variables (e.g., volume, intensity, load, and frequency) are key (Kraemer & Ratamess, 2004). Manipulations to training intensity and volume have received most of the attention but training frequency has largely been overlooked (Grgic et al., 2018; Ralston et al., 2017).

The role of training frequency has been debated, and the optimal frequency is not clear. Training frequency is defined in the literature as the number of training sessions performed for a given period, usually described on a weekly basis (Kraemer & Ratamess, 2004). Frequency has be further characterized by the number of training sessions per week per muscle group or exercise (Schoenfeld et al., 2015), which is the definition used in this article. The American College of Sports Medicine (2009), recommends that novices and untrained individuals should train every muscle group 2–3 times per week (1). However, this recommendation of training frequency has been the subject of some criticism since it is based on limited evidence (Grgic et al., 2018; Ralston et al., 2018; Schoenfeld, Grgic & Krieger, 2019). As a result, there has been a small renaissance on training frequency, with multiple studies published on the topic. One study that has received much attention was the “Norwegian Frequency Project,” which showed positive results favoring higher frequency training for elite/trained powerlifters (Raastad et al., 2012). The problem with this study is that it was only used as a conference paper and never published in a journal, so it is difficult to control and verify the methods used in the project.

Although the number of studies published is increasing, the total pool of studies is still limited. To the best of our knowledge, there are eight published studies that explore the effects of training frequencies on muscle adaptations on trained males under equal volume conditions (Brigatto et al., 2019; Colquhoun et al., 2018; Gentil et al., 2018; Gomes et al., 2019; Lasevicius et al., 2019; Mclester, Bishop & Guilliams, 2000; Saric et al., 2019; Schoenfeld et al., 2015). Several of these studies have focused on lower training frequencies, that is, three or lower. Only three of these studies controlled for the effect of training frequencies higher than three (Colquhoun et al., 2018; Gomes et al., 2019; Saric et al., 2019). Furthermore, most of these studies did not find any differences in gains in 1RM between training frequencies except Mclester, Bishop & Guilliams (2000), who reported that the gains in 1RM by training once a week were 33% lower than training three times per week. However, in this study the volume was very low compared with the other studies and men and women were combined, which could influence the results.

Two recent meta-analyses noted that the literature on training frequency under equal volume conditions is small and suggested that future research is needed (Grgic et al., 2018; Ralston et al., 2018). Ralston et al. (2018) noted that studies with trained subjects were needed. Furthermore, Dankel et al. (2017) suggested that an increase in training frequency could be advantageous to spread the total training volume to counteract muscle fatigue and overtraining. Training with a very high volume in one training session can induce high levels of fatigue and prolonged recovery time, which can be suboptimal for athletes that try to induce specific neuromuscular adaptations (Pareja-Blanco et al., 2018). Seen in the context of motor learning theory, it also can be assumed that more frequent training of a movement could lead to a higher increase in strength, due to an improvement in neural efficiency (Shea et al., 2000).

Since it is unclear whether exercise frequency affects muscular strength under equal total exercise volume, especially in higher training frequencies (Grgic et al., 2018), the purpose of this article was two-fold: first, to investigate the effect of training frequency of two vs four times per week when matched on total training volume upon maximal muscle strength in strength-trained males; second, to investigate the effects of training frequency on perceptual responses (rate of perceived exertion) among the subjects. We hypothesized that training with a frequency of four sessions per week would promote greater increases in maximal strength with a lower self-reported rate of perceived exertion (RPE) compared to two sessions per week due to the lower workload per muscle group per training session.

Methods

Subjects

Subjects were 21 male volunteers (height: 1.85 ± 0.06 m, body mass: 85.3 ± 12.3 kg, age: 27.6 ± 7.6 years) who were recruited subjects that attend the local gym. The inclusion criteria were the subject had to be male, could be defined as trained (a least 1 year experience of resistance training with a minimum of two workouts per week) with experience training on barbell back squat and bench press, was free of injuries, and stated they had not taken any performance enhancing drugs. The mean resistance training age of the group was 4.7 ± 2.8 years. Each subject was informed of the testing protocol, training procedures, and possible risks; and written consent was obtained from the subjects prior to the study. The study was conducted with the approval of the Norwegian Center for Research Data project number: 42440 and conformed to the latest revision of the Declaration of Helsinki.

Study design

To investigate the effect of training frequency with the same training volume upon strength (1 repetition maximum in the bench press and squats) and RPE, a pretest-posttest randomized group design was used. Subjects were randomly assigned to one of two experimental groups: a SPLIT group where the training protocol was divided into two sessions training barbell back squat and exercises for the lower body and two sessions training bench press and exercises for the upper body; or a FULLBODY group where subjects trained four full-body sessions with barbell back squat and bench press each time, together with four other supplementary exercises for the whole body. A summary of the resistance training protocol can be found in Table 1. Throughout the 8-week training period, all resistance training variables were held constant, especially total training volume (repetitions × set × intensity), between the two conditions, except the training frequency. The training protocol was built up with a pretest the week before the training period and a posttest the week after.

Table 1 Schematic overview of the training protocol.

Protocol	Day 1	Sets	Day 2	Sets	Day 3	Sets	Day 4	Sets	
SPLIT	Bench press	6	Back squat	6	Bench press	5	Back squat	5	
	Bent over row	3	Stiff legged deadlift	3	Lat pulldown	3	Leg press	3	
	One arm dumbbell row	3	Lunges	3	Seated cable row	3	Leg curl	3	
	Overhead press	3	Leg extension	3	Lateral raises	3	Calf raises	3	
	Biceps curl with dumbbells	3			Standing cable triceps curl	3			
	Face pulls	3			Face pulls	3			
FULLBODY	Back squat	3	Bench press	3	Back squat	3	Bench press	3	
	Bench press	3	Back squat	3	Bench press	2	Back squat	2	
	Bent over row	3	Seated cable row	3	One arm dumbbell row	3	Lat pulldown	3	
	Leg curl	3	Leg extension	3	Leg press	3	Stiff-leg deadlift	3	
	Biceps	3	Overhead	3	Triceps	3	Lunges	3	
	Face pulls	3	Calf raises	3	Face pulls	3	Lateral raises	3	

Procedures

One week before and after the intervention period, maximal strength was assessed by a one-repetition maximum (1RM) test in barbell back squat (1RMSQUAT) and bench press (1RMBENCHPRESS). The 1RM test was done following the guidelines established by the National Strength and Conditioning Association (Haff, Triplett & National Strength & Conditioning Association (US), 2016). The subjects started with a 5–10-min general warm-up consisting of running on a treadmill, followed by a set of five repetitions at around 50% of an estimated 1RM and 2–3 sets of 2–3 repetitions around 60–80% of the estimated 1RM. The subjects then performed one repetition sets with increasing load to establish their 1RM. They had a maximum of five attempts to determine the 1RM. 1RM in barbell back squat was always tested first followed by testing 1RM in bench press. Between each successful attempt, the subject rested for 3–5 min before the next set with increased weight. To get an attempt approved in the barbell back squat, the subjects had to meet the parallel depth and a green light from the test leader. In bench press, the subject had to have head, shoulders, and bottoms placed on the bench and the feet placed on the floor during the lift. They had to lower the barbell to their chest and had to achieve full extension in the elbow to get the lift approved. The subjects were asked to refrain from any other exercise for 24 h before testing.

A rating of perceived exertion (RPE) was used to test if there was a difference between the two groups after the exercises and workouts. RPE scales have been well-established as methods of determining exertion during exercise (Helms et al., 2016). The Borg CR10 scale was used as the RPE scale to quantify the perception of physical exertion (Morishita et al., 2013). After completing each of the exercises (barbell back squat and bench press), and a couple of minutes after the exercise session, the subject was instructed to rate their perceived exertion by choosing a number on the CR10 scale. A rating of 0 was categorized as no exertion or at rest; a rating of 10 was the maximal exertion they can achieve. The RPE measurement for each of the three rating points was analyzed as a weekly mean for each of the subjects. All ratings through the training week for each of the three measurements were summed and divided by the total number of measurements in that week. The subjects were familiar with using the CR10 scale since they regularly evaluated their training intensity by using this scale.

After the pretest, the subjects were randomly assigned to one of the two experimental groups: SPLIT (n = 10, height: 1.84 ± 0.05 m, body mass: 87.0 ± 13.3 kg, age: 30.6 ± 9.5 years) and FULLBODY (n = 11, height: 1.87 ± 0.07 m, body mass: 83.7 ± 11.6 kg, age: 24.8 ± 4.0 years); each group trained under matched volumes. To control for volume, the total weekly resistance training volume (repetitions × set × intensity) was equated between the groups. The volume was equated because a dose-response relationship between volume and increase in muscular strength has been previously reported (Heaselgrave et al., 2019; Ralston et al., 2018; Rhea et al., 2003). The SPLIT training group trained with a frequency of two sessions per muscle group; the training protocol was divided into two lower-body and two upper-body workouts. The FULLBODY training group had a training frequency of four sessions per muscle group; they trained four full-body workouts per week.

The training protocol for both groups also included a mixture of single- and multi-joint exercises for the rest of the body (Table 1). The weekly total training volume of the two groups was equal. Back squat and bench press had the training intensity determined based on their one-repetition maximum (1RM) and was a percentage of the 1RM. The repetition ranges for these exercises were between two and ten, with the higher repetitions in the first of weeks. The intensity was between 65% and 90% of their 1RM. The intensity and repetition ranges for the bench press and back squat was periodized with more repetition and lower intensity during the first weeks; throughout the training period, the repetitions and intensity gradually shifted to fewer repetitions with high intensity. Weekly repetition ranges and intensity is reported in Table 2. The weekly set volume of these exercises was set to 11, which follows the recommendation from Ralston et al. (2017), who recommended a medium to high weekly set count for maximizing strength gain.

Table 2 Schematic overview of the load and repetitions ranges per week.

Week	Weekly Sets	Back Squat	Bench press	
Rep range	Intensity (% of 1RM)	Rep range	Intensity (% of 1RM)	
1	11	5–7	70–75%	7–10	65–70%	
2	11	3–7	72.5–80%	6–10	67–72.5%	
3	11	3–6	75–80%	6–10	70–75%	
4	11	2–5	77.5–87.5%	4–8	72.5–80%	
5	11	3–5	75–82.5%	3–5	72.5–82.5%	
6	11	3–5	75–85%	3–5	77.5–85%	
7	11	2–4	77.5–87.5%	2–5	77.5–87.5%	
8	11	2–5	77.5–90%	2–4	80–90%	

In the other exercises, the repetition ranges were between eight and twelve repetitions, with the intensity set to two repetitions in reserve, that is, is the number of repetitions the athlete feels he has left in the tank. This means that with the two repetitions in reserve, the subject had to feel that he had a maximum of two repetitions left to fatigue after a set of one exercise on that weight (Zourdos et al., 2016). If the subject felt that he could do more, he was instructed to increase the weight by 2–10% on the next set. The two repetitions in reserve method for the supplementary exercises was used to have the possibility for autoregulation within each subject and to avoid that the subject performed until full exhaustion which could influence training the main exercises.

Statistical analyses

All statistical analyses were performed using SPSS 25.0 for Windows (SPSS Inc., Chicago, IL, USA). The normality and homogeneity of the variances were verified using the Shapiro–Wilk and Levene’s tests. Descriptive statistics (mean ± SD) were calculated for each dependent variable for the pretest and posttest (1RMSQUAT, 1RMBENCHPRESS). To compare the effect of the two protocols, a two-way ANOVA of 2 (pretest and posttest) × 2 (groups: FULLBODY and SPLIT) was performed for each of the strength assessments. The percentage increase was also calculated for 1RMSQUAT and 1RMBENCHPRESS.

A two-way ANOVA of 2 (groups: FULLBODY and SPLIT) × 8 (weekly mean of RPE, week 1 to 8) was performed for the three different RPEs (RPE after bench press, squat, and workout). Assumptions of sphericity were evaluated using Mauchly’s test; where sphericity was violated (p < 0.05), the Greenhouse–Geisser correction factor was applied. A one-way ANOVA (weekly mean of RPE, week 1 to 8) was also done per group for the three different RPEs to identify the development per group. When significant differences occurred, Holm–Bonferroni post hoc tests were conducted to identify statistically significant comparisons. The level of significance was set at p < 0.05, and all data were expressed as mean ± SD. Effect size was evaluated with η2 (eta squared) where 0.01 < η2 < 0.06 constitutes a small effect, 0.06 < η2 < 0.14 constitutes a medium effect, and η2 > 0.14 constitutes a large effect (Cohen, 1988).

Results

At baseline, no significant difference in 1RM in squat (p = 0.55) and bench press (p = 0.46) were found between the groups. Each group increased in the bench press (SPLIT group: +7.75 kg; FULLBODY group: +8.86 kg; F = 223.9, p < 0.001, η2 = 0.92) and squat exercise (SPLIT group: +13.25 kg; FULLBODY group: +14.31 kg; F = 152.9, p < 0.001, η2 = 0.89) significantly from pre to post test (Fig. 1). Relatively, the SPLIT and FULLBODY training groups increased by 7.7% and 9.7%, respectively, in 1RM bench press and by 12.1% and 11.5%, respectively, in 1RM squats. The relative gain in 1RM in squats was significantly higher than in 1RM in bench press (p = 0.022). The was no effect of group (F ≤ 0.71, p ≥ 0.40, η2 ≤ 0.036) or interaction effect of group*time (F ≤ 1.0, p ≥ 0.33, η2 ≤ 0.05) for strength (Fig. 2).

Figure 1 Mean 1RM of squat and bench press (±SD) at pre- and posttest for SPLIT and FULLBODY training groups.

Mean 1RM of (A) squat and (B) bench press (±SD) at pre- and posttest for each subject and the average of SPLIT and FULLBODY training groups. An asterisk (*) indicates a significant increase in 1RM from pretest for this group at p < 0.05.

Figure 2 Absolute individual change from pretest to posttest in 1RM squat and bench press performance with average per group (dotted line).

Absolute individual change from pretest to posttest in (A) 1RM squat and (B) bench press performance with average per group (dotted line).

The rating of perceived exertion (RPE) for the exercise bout (F = 4.9, p < 0.001, η2 = 0.21), after the squat (F = 10.2, p < 0.001, η2 = 0.35) and bench press (F = 3.0, p = 0.043, partial n2 = 0.14) exercises per week were influenced during the intervention period. Only a significant time*group effect was found in RPE after the squat exercise (F = 3.5, p = 0.002, η2 = 0.15). In addition, a non-significant but large effect was found between the two training groups in RPE after the squat exercise (F = 3.34, p = 0.083, η2 = 0.15).

The post hoc comparison revealed that the subjects in the split group reported significantly higher session RPE in week 4 compared with week 3 and 5; the RPE increased again from week 6 to 8 (Fig. 3). In the FULLBODY group, the session RPE decreased from week 1 to 2 followed by an increase until week 4. In week 5, it decreased again significantly (Fig. 3).

Figure 3 Average (±SD) rating of perceived exertion for whole training bout, after barbell back squat, and bench press per week.

Average (±SD) rating of perceived exertion for (A) whole training bout, after (B) barbell back squat, and (C) bench press per week. An asterisk (*) indicates a significant difference between the two groups for this week at p < 0.05; → indicates a significant difference from this RPE with the next one at p < 0.05.

Rating of perceived exertion after the squat was significantly higher in the SPLIT group in weeks 4 and 5 compared with the FULLBODY group. Also, the development of the RPE per week after squats followed a different development: while the RPE increased in week 4, decreased in week 5, and increased again the last two weeks for the SPLIT group, the RPE of the FULLBODY group decreased to a minimum in week 5, after which it increased again in week 6 (Fig. 3).

Rating of perceived exertion after bench press only changed in week 2, in which only the FULLBODY group had a significantly lower RPE compared with the other weeks during training. This also resulted in a significant difference with the SPLIT group in week 2 (Fig. 3).

Discussion

The main aim of this study was to investigate the effect of resistance training frequency on maximal muscular strength and RPE by training twice vs four times a week when matched on total training volume. The main findings were that both training frequencies achieved a similar significant increase in maximal strength (1RM) in the barbell back squat and bench press over the 8 weeks of training. However, RPE developed differently during the training period in which, especially after the squat exercise, RPE seems to be higher some weeks for the SPLIT group compared with the FULLBODY group.

Both SPLIT and FULLBODY groups had a similar increase in strength from pretest to posttest in both 1RMSQUAT (13.25 and 12.27 kg, respectively) and 1RMBENCHPRESS (7.75 and 8.86 kg, respectively), which indicates that 8 weeks of training, regardless of frequency, will increase muscle strength, as long as the weekly training volume in the exercises, barbell back squat, and bench press are high enough. The result of this study follows the trends shown in other studies (Brigatto et al., 2019; Colquhoun et al., 2018; Gentil et al., 2018; Gomes et al., 2019; Lasevicius et al., 2019; Saric et al., 2019; Schoenfeld et al., 2015) on the topic, with the effect of an increase in frequency not yielding a significantly greater effect on maximal strength. Only Mclester, Bishop & Guilliams (2000) reported that a lower frequency group achieved only 2/3 of the increase in strength of the high-frequency group, but they compared one session per week with three sessions per week.

Regardless of frequency, the relative 1RM gain in squats was higher than in the bench press. This can be explained by two mechanisms. The first explanation could be in the difference in loading schemes for the exercises. The protocol for bench press had a lower percentage of 1RM in the first couple of weeks, which could have been a less optimal scheme than the scheme for the barbell back squat. The second explanation could have been the higher set-volume on the muscles in the legs by the “support” exercises prescribed in the protocol. The protocol prescribed both multi-joint, such as lunges, and single-joint exercises, such as leg extensions, focused on the legs (Table 1). Some researchers argue these have to be counted in the weekly sets on the muscles (Schoenfeld et al., 2019). The chest muscles were only trained by the bench press, with three weekly sets of a triceps exercise as a “support exercise.”

Although the current findings suggest that exercise frequency does not have an overall effect by itself on muscle strength, it can be an important variable to consider when developing training programs. As the level of athletes increases, manipulation of training variables becomes more important (Kraemer & Ratamess, 2004). One of the methods to ensure further adaptation for athletes when the training level increases, is to correspondingly increase the total weekly volume. This can be done in different ways, such as increasing weekly sets, repetitions per set, and load (Kraemer & Ratamess, 2004). When the total weekly sets for an athlete reaches an upper limit, it could be advantageous to spread volume over several training sessions, as suggested by Hartmann et al. (2007), to reduce the likelihood of overtraining. Exercising at too high of a volume per session can be less effective at maximizing muscle adaptations. There is a limit to the number of good quality sets due to fatigue (Boyas & Guevel, 2011), but this threshold is different for each individual. Some studies have shown favorable outcomes to strength when training at a lower number of sets per session is introduced (Amirthalingam et al., 2017). Amirthalingam et al. (2017) concluded that exercising at 4–6 sets per muscle group within a workout was optimal for muscular adaptations and increasing the number of sets within a session to greater than this number did not appear to produce a greater effect. An increase in total training volume (repetitions × set × intensity) in one session and, therefore, nearer to failure has also been shown to significantly increase the recovery time needed (Pareja-Blanco et al., 2018). In our study, a recovery time effect did not occur, since the total training volume was at a medium level (Weekly sets were at 11) and the intra-session sets were also low, with 5–6 sets per session for the SPLIT group and 2–3 for the FULLBODY group. This effect could be the reason why the Norwegian Frequency Project showed positive effects of higher training frequency (Raastad et al., 2012), because higher level/elite athletes need a higher weekly set volume to get adaptations. However, this is speculation, because the methods of that study cannot be verified or controlled. There is a possibility that frequency can have an effect when weekly sets are very high, but further research is required to develop an understanding of this; as of this publication (Raastad et al., 2012), no studies have been conducted on very high weekly sets (i.e., more than 20 sets).

The present findings also contradict motor learning theory, that is, that practicing an exercise more frequently will induce higher strength gains due to higher improvement in neural efficiency (Shea et al., 2000). Our findings follow the hypothesis of Sale (1988) that this effect is limited for trained subjects. Our results demonstrate that practicing a strength exercise twice a week could be proficient to increase neural efficiency for trained subjects.

Fatigue could influence strength gain and recovery of athletes due to the total training volume per muscle group per session. To this end, the RPE can be an important tool for resistance-trained subjects and coaches during exercise execution or training sessions (Foster et al., 2001). We found that subjects reported RPE changes during the training period per week in all three measurements (RPE after the training bout, barbell back squat, and bench press), which is an indication that the intensity changed throughout the training period, especially around week 4 and 5. This follows previous studies that have shown a correlation between the reported RPE and the intensity prescribed for 1RM under resistance training (Naclerio et al., 2011; Pincivero, Coelho & Campy, 2003). The main changes in RPE per week occurred after the barbell back squat with the indication that the subjects reported higher RPE after the squat for the SPLIT group then the FULLBODY group (Fig. 3) throughout the training period. This difference between the exercises RPE indicates that a higher number of sets with barbell back squats in one session can induce higher fatigue; splitting the total sets of barbell back squat into two sessions can be favorable for perceived exertion. This result could also be an effect of the number of sets done with “support exercise” and single-joint exercises in the training protocol. The increased number of sets done on the muscles that are used can increase the subjects’ fatigue and increase the recovery time needed after the training session.

This study had several limitations. First, the study only lasted 8 weeks. Although the duration was sufficient to achieve a significant increase in strength for both barbell back squat and bench press, over a longer training time, differences between the groups could occur. Second, the small sample size affected the statistical power, as most longitudinal studies in this field. Third, the results are specific to resistance-trained men. Men and women could have a difference in fatigability (16). It has been suggested that women have a quicker recovery rate on muscle fatigue than men after resistance training (Judge & Burke, 2010), and therefore may experience better effects of higher training frequency than men. However, to our knowledge, there are no studies testing the effects of training frequency on trained women. Fourth, this study did not control for the dietary intakes of the subjects. The subjects may not have had an optimal nutritional intake during the training period, which may affect the results; however, the randomization of the subjects should have prevented such a bias.

Based on our findings, we conclude that training with a frequency of two and four sessions per muscle group are both viable approaches to increase muscle strength in the barbell back squat and bench press for trained males, as long as the total weekly training volume is equal. It is possible that spreading the weekly volume to different days could be favorable for the rating of perceived exertion, especially for exercising the muscles in the lower body. The group with a training frequency of four reported a lower RPE for barbell back squat than the group training with a frequency of two. This study corresponds with previous studies and with two meta-analyses on frequency training; it seems that the effect of increasing the training frequency does not have an equally important role as volume and load on strength gains (Grgic et al., 2018; Ralston et al., 2018).

Conclusion

The results of this study suggest that both training with a frequency of two and four times per week provides similar increases in maximal strength for trained subjects under the same total weekly volume. The RPE result in this study suggests that it could be favorable to spread the total training volume in several training bouts throughout the week, especially for training the muscles in the lower body. This suggests that higher training frequencies could be used as a tool to counteract perceived exertion for athletes since the training volume per session will be lower. Both results give coaches and athletes greater variety in how to structure a training program with different training frequencies without sacrificing an increase in performance. Programs can then be periodized with different training frequencies to follow the athlete’s personal preferences, time constraints, or when the daily training volume is no longer manageable.

Supplemental Information

Supplemental Information 1 Raw data.

Click here for additional data file.

Additional Information and Declarations

Competing Interests

Author Contributions

Human Ethics

Data Availability

The authors declare that they have no competing interests.

Emil Johnsen conceived and designed the experiments, performed the experiments, analyzed the data, prepared figures and/or tables, authored or reviewed drafts of the paper, and approved the final draft.

Roland van den Tillaar conceived and designed the experiments, analyzed the data, prepared figures and/or tables, authored or reviewed drafts of the paper, and approved the final draft.

The following information was supplied relating to ethical approvals (i.e., approving body and any reference numbers):

The Norwegian Center for Research Data approved this research (project number: 42440).

The following information was supplied regarding data availability:

Raw data are available as a Supplemental File.

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
