# Peer review of "Effects of training frequency on muscular strength for trained men under volume matched conditions"

_PeerJ, doi:10.7717/peerj.10781_

## Round 0.1 · original submission · Major Revisions

The topic is interesting. The paper can be improved following the suggestions of the reviewers.

Reviewer 1 ·

Basic reporting

There are multiple errors of writing in the document.
References seem suitable but improvement on data reporting and visual depiction required.
Specific comments have been provided to authors.

Experimental design

Study fits aims and scope of journal.
Research question is interesting but limited conclusions can be drawn based on lack of outcome measures and several issues with the study design. Specific comments provided in latter section and how this can be improved.

Validity of the findings

Some contradictory statements when interpreting the results. Some statements made in discussion require further support or need to be checked carefully against existing literature.
The study has meaning for the strength and conditioning community but lack some novelty due to limited information that can be drawn from the results.
Specific comments have been provided to the authors on how some of these points could have been addressed.

Additional comments

The authors present a study where resistance training frequency is conducted over 8 weeks in resistance trained males. Groups consisted of a split or full-body program meaning that main exercises were trained either 2 or 4 days per week, respectively, under volume matched conditions. The main findings were that there are no additional benefits of increased training frequency on muscular strength adaptations however, this may be favourable to avoid increases in ratings of perceived exertion. I have provided specific and more general comments below that I hope the authors will find useful:

Specific comments:

L31: ‘tested prior to and after’
L35: The supplementary exercises listed in the table are not full body. Please amend.
L39-42: This section would benefit from presenting some of the raw data not just p-values. Moreover, no numerical or statistical results are presented for RPE.
L46: The conclusion about RPE is not clear based on the rest of the abstract.
L51: Here the authors say ‘recently’ but strength training has been popular for many years. In fact, the study referenced here is 13 years old.
L53-54: Please check wording in this statement.
L59: ‘but training frequency’
L66: space required after reference. And should this be American College of Sports Medicine?
L70-71: Again the ‘last few years’ are referred to but the unpublished study in question dates back to 2012.
L70: ‘One study that has received’
L75-76: This statement contradicts itself.
L106: ‘local gym participants; is unclear. Do you mean people who attended gyms?
L109: enhancing
L11: ‘and written consent’
L117: ‘in the bench press’ and ‘squat’
L133: ‘repetitions at’
L192: 1RMSQUAT is spelt incorrectly in text.
L246: ‘in other studies’

There are also several writing errors in the discussion and conclusion. I invite the authors to check the whole document carefully to improve the overall quality.


General comments:

L75-82: here it would be good to present a little more about the specific study details and main findings.
L105: The heading is ‘participants’ but in text ‘subjects’ is used. Please check and keep consistent in all sections.

Introduction: The rationale for measuring RPE is not well supported in my opinion. For example, many other simple questionnaires would give perhaps more substantial information rather than just information about the session itself e.g. perceived fatigue after or prior to a session, motivation to train, muscle soreness questionnaires etc. This would provide more detail to the findings and enable a more informed decision about subjective outcomes and training frequency to be made by the authors.

Study design: Here it would be good to either list the supplementary exercises, or refer the reader to the table. The exercises actually occurred was not clear until the table was inspected.

L149-150: Here the terms ‘exertion’ and ‘effort’ are used interchangeably. Technically, these can represent different things so please revise and clarify what the focus was on, i.e. effort or exertion.

Pre-training statistical results for between group characteristics at baseline should be presented to ensure that they were not different.

L176-177: It is unclear why the repetition in reserve method was used for supplementary exercises? As some of these muscles are synergists etc in the main movements this prescription could affect the overall results, recovery times between session etc. Further clarity on this decision is required.

Results: In text it would be useful to present some raw values rather than just statistical outcomes.

The results of RPE and interpretation of this throughout the manuscript is difficult to follow. For example, on L223-224 it states that RPE was higher in the split group (weeks 4-5) but the conclusion of the abstract seems to state that splitting the session avoids increases in RPE? Please also check for consistency in other sections.

L251-253: The is a very strong statement that is not well supported. Suggest removing or discussing further with additional supporting references.

L255: The idea of relative and absolute strength here seems to come out of nowhere as it is not presented in the results section.

L280: Suggest some caution a sit appears Pareja Blanco reports that training to failure, especially with higher repetition schemes delays recovery. Currently, the manuscript seems to imply that increased volume means closer to failure which is not technically true as it depends on the intensity used.

L315: Saying differences between groups could increase implies that there may have been a difference to start with, but the results don’t seem to support this.

L341: RPE is not specifically about fatigue so this sentence needs to be revised.

L333: In the introduction the two meta-analyses are listed as Ralston and Grgic. Here Rhea is listed, is this correct?

References 421-423: The full stops appear to be in the incorrect place.

Table 1: The exercises described as ‘biceps’ and ‘triceps’ do not provide sufficient information. Please provide more specific detail about the exercises.

Table 2: So here the repetitions and intensity ranges can vary even within each week. This presents a problem when the aim of the study was to ‘volume-match’. So quite possibly the performed repetitions and loads between groups could have been different. Moreover, how was it decided upon what intensity and the number performed by the individual for each training session. At the very least, training data showing intensities and volumes performed by each group per week, and statistical analysis of potential differences is required.

Figure 1: It is better to show the results as individual data points with a possible line to indicate individual changes over the training intervention. For example, it may reveal that the weakest individuals to start with improved strength the most. This leads on to figure 2 where individual improvements are shown but provides no insight into the starting strength. I suggest that the presentation of this data is considered to improve transparency and provide further interesting insight.

·

Basic reporting

There are commas missing for many of the in-text citations (eg: L63 (Kraemer & Ratamess 2004) )

L25 Change to something along the lines of “In strength training, the role of training frequency is often debated but the limited data available does not allow for clear training frequency “optimisation” recommendations"

L66 add a comma after (2009)

L72 Well done on clarifying that the “Norwegian Frequency Project” has not been fully published therefore limiting the applicability of its findings

L120 Exercise needs to be plural (right before “for the upper body”)

Experimental design

L107 Was there a particular reason to include only males in this study? If yes, please clarify

L136 Were the 1RM testing procedures based on any studies/guidelines? What was the rationale behind allowing participants to have 5 attempts in order to determine their 1RM?

L149 Were the participants familiar with the CR10 scale? If not, did they have to go through a “familiarisation” period prior to being tested? Please clarify.

Validity of the findings

No comment

Additional comments

The authors have submitted a very interesting and well-presented study exploring the effects of training frequency on the 1RM strength of trained males. This study is a great addition to the, currently limited, strength training literature.

Reviewer 3 ·

Basic reporting

no comment

Experimental design

1RM – The participants were trained in resistance training, but why did not perform test and re-test of 1RM in back squat and chest press?

RPE

Line 148 - When was the RPE asked at the end of the exercise session? Right after it is over or a few minutes later? Please clarify?

The Borg CR10 scale was trained with the participants before the data collection?
Were the participants familiar with the Borg scale?

Line 158 – The time under tension was controlled to calculate the volume?

Validity of the findings

no comment

Additional comments

This article deals with an important and relevant topic for coaches, personal trainers, athletes and resistance training practisers and reveals great importance in terms of practical application.
I suggest some changes in order to enhance the quality of the paper further.

Abstract

Line 25 – introduce “to increase maximal strength”
Line 27 – I suggest introduce “maximal muscular strength”

Introduction

In the text, the authors use different nomenclatures to refer to the training to increase muscle strength (strength training, resistance training). Please use only one.

Line 236 – Introduce “maximal muscular strength”
Line 249 – Introduce “maximal strength”

There is different muscular strength manifestation. In this paper, it was evaluated the maximal strength and not the other manifestation forms. Please be more specific when made the discussion/comparisons with other studies.

---

## Round 0.2 · Minor Revisions

The current version of the manuscript is improved in quality and clarity, hovewer there are still comments which should be addressed plus a few additional errors.

Reviewer 1 ·

Basic reporting

This is an improved version of the manuscript. There are still some writing errors and several amendments that require attention.

Experimental design

N/A

Validity of the findings

The data and conclusions are clearly in this improved version of the manuscript.

Additional comments

The authors have worked to ensure that the current version of the manuscript is improved and as such I believe that the quality and clarity has also improved. I still have several comments which should be addressed plus a few additional errors that I picked up. See below.

Figure 1. Please make the pre and post point a circle for each individual and the line style the same across all individuals.
Table 1. Bicep and tricep exercises still unclear. I.e. dumbbell, barbell, cable, seated, standing, unilateral etc?
Table 2 caption: Please clarify that this volume was the same during 'each week' not 'one week'. The authors should also clearly say that this applied to the SQ and BP exercise specifically.

Abstract: Opening statement is long and does not read well.
L49: 'avoid potential increases in RPE'
L66: 'has'
Intro general: L84 states that there are multiple studies published on the topic but earlier the authors imply that frequency has largely been overlooked. I suggest be more clear and consistent in several sections of the introduction.
L288: Furthermore, Dankel et al...
L295: A reference was removed from here but I feel it is required.
L307: It may be better to refer to 'perceptual responses' rather than felt fatigue as this is not captured by RPE.
L350: Reference required.
L414L Prefer to avoid abbreviation as it is only used a few times. Too many abbreviations can become confusing.
L444: close bracket required after p=0.46.
L515: suggestion: 'Although the current findings suggest that exercise frequency...'
L557: I am not sure it is correct to discuss the unpublished study here.
l564: Reference required.
l569: RPE
L597-599: Wording here is not easy to follow. Please amend.
L604: 'higher training frequencies than men.'
L632: 'provides similar'
L635: Clarify to the reader that this higher training frequency means that volume per session is lower (in equated models). This is probably what provides the benefit in this context based on your current study.

·

Basic reporting

No further comments. The authors have addressed my previous comments.

Experimental design

No further comments. The authors have addressed my previous comments.

Validity of the findings

No comment.

---

## Round 0.3 · accepted · Accept

The current version of the manuscript is improved and clearly readable.